# Droplet Digital PCR or Real-Time PCR as a Method for Quantifying SARS-CoV-2 RNA in Plasma—Is There a Difference?

**DOI:** 10.3390/v17060772

**Published:** 2025-05-28

**Authors:** Beathe Kiland Granerud, Mari Kaarbø, Huda Al-Baldawi, Kari Otterdal, Bente Halvorsen, Andreas Lind, Simon Rayner, Jan Cato Holter, Susanne Dudman

**Affiliations:** 1Institute of Clinical Medicine, University of Oslo, 0318 Oslo, Norway; huda.al-baldawi@medisin.uio.no (H.A.-B.); b.e.halvorsen@medisin.uio.no (B.H.); simon.rayner@medisin.uio.no (S.R.); jacaho@ous-hf.no (J.C.H.); 2Department of Microbiology, Oslo University Hospital, 0450 Oslo, Norway; mari.kaarbo@medisin.uio.no (M.K.); uxlndr@ous-hf.no (A.L.); 3Institute of Nursing, Health and Laboratory Science, University College of Østfold, 1671 Kråkerøy, Norway; 4Research Institute for Internal Medicine, Oslo University Hospital Rikshospitalet, 0372 Oslo, Norway; kari.otterdal@ous-research.no

**Keywords:** SARS-CoV-2, RNAemia, qRT-PCR, RT-ddPCR, plasma, COVID-19, E-gene

## Abstract

The aim of this study is to ascertain whether qRT-PCR (reverse transcriptase real-time PCR) or RT-ddPCR (reverse transcriptase digital droplet PCR) is more effective for detecting SARS-CoV-2 RNA (severe acute respiratory syndrome coronavirus 2 RNA) in blood plasma from COVID-19 (coronavirus infectious disease-19) patients. The E-gene of SARS-CoV-2 RNA was quantified using both methods in 128 plasma samples from 70 hospitalized patients, followed by a statistical analysis to compare the sensitivity and concordance between the methods. Out of the 128 samples, 89 yielded consistent results irrespective of the method used, whereas 39 samples showed discrepancies between the two different methods. RT-ddPCR frequently registered higher viral quantities compared to qRT-PCR; however, the results did not demonstrate a clear superiority in sensitivity for RT-ddPCR. Although RT-ddPCR registered higher viral quantities, this study concludes that both methods provide comparable results for detecting SARS-CoV-2 E-gene RNA in plasma.

## 1. Introduction

SARS-CoV-2 (severe acute respiratory syndrome coronavirus 2) not only causes mild-to-severe respiratory illness but also has extrapulmonary manifestations [1]. The virus primarily employs ACE2 (angiotensin-converting enzyme-2) and its co-receptor TMPRSS2 (transmembrane protease serine 2) for cellular entry [2]. However, ACE2-independent entry has also been suggested [3,4], potentially linked to naturally occurring mutations in the spike protein [5]. Following infection of the lung epithelium, SARS-CoV-2 may cross over to the endothelium [6,7]. SARS-CoV-2 RNAemia (the presence of SARS-CoV-2 RNA in the bloodstream) correlates with markers of tissue damage in endothelial cells or lung epithelium [8] and viral replication within the lungs [9,10]. Additionally, circulating SARS-CoV-2 virus particles (viraemia) have been detected in plasma, although their infectivity remains uncertain [11]. Despite the low expression levels of ACE2 in blood [12], research has demonstrated that SARS-CoV-2 can infect and, to some extent, replicate in monocytes, B lymphocytes, and T lymphocytes, possibly via ACE2-independent mechanisms [13,14,15,16]. Nevertheless, these findings may also in part be attributable to the viral particles adhering to the cell surface. Additionally, nuclease resistance and protection by cell membranes may also explain the persistent presence of SARS-CoV-2 in blood [17].

Numerous studies have identified an association between SARS-CoV-2 RNAemia and increased morbidity and mortality in COVID-19 (coronavirus disease 2019) patients [9,11,18,19,20,21,22,23,24,25,26,27,28,29,30,31,32,33,34,35,36]. Another observed association is the persistence of SARS-CoV-2 RNA in blood and the development of post-COVID-19 syndrome [37], also known as long COVID. Thus, the detection of SARS-CoV-2 RNAemia could potentially serve as a prognostic biomarker for the prolonged severity of COVID-19.

Detection of RNAemia is typically conducted via one-step qRT-PCR (quantitative reverse transcription PCR) or one-step RT-ddPCR (reverse transcription digital droplet PCR). The similarities and differences of these two methods have been thoroughly described by Quan et al. [38]. In short, RT-ddPCR enables absolute quantification while qRT-PCR relies on relative quantification. Also, RT-ddPCR is less affected by inhibitors than qRT-PCR and is believed to be a more sensitive assay but is more technically challenging and time-consuming.

Most studies on SARS-CoV-2 RNAemia utilize qRT-PCR, with only a few adopting RT-ddPCR [18,19,22,30]. The method’s effectiveness in detecting RNAemia may influence clinicians’ decisions regarding disease prognosis and the need for closer follow-up after hospital discharge.

Studies comparing RT-ddPCR and qRT-PCR for SARS-CoV-2 detection have thus far used RNA extracted from upper respiratory swabs, synthetic RNA standards, or cultured virus, rather than RNA extracted from blood plasma [39,40,41,42,43,44,45,46,47,48,49,50]. Only a limited number of studies target the E-gene for PCR amplification [42,45,48,50]. Therefore, our aim was to evaluate the qualitative test performance of qRT-PCR versus RT-ddPCR for detecting SARS-CoV-2 RNA in plasma, using the E-gene oligomer [51].

## 2. Materials and Methods

### 2.1. Sample Material

EDTA plasma was sourced from the Norwegian SARS-CoV-2 study (NCT04381819), a prospective, observational, multicenter study. The study participants were inpatients recruited from five different Norwegian hospitals from 10 March 2020 to 1 September 2021. Peripheral venous blood samples were collected in pyrogen-free blood collection tubes with K_2_EDTA anticoagulant, immediately cooled on melting ice, and centrifuged within 30 min at 2500× *g* for 20 min at 4 °C. The plasma was aliquoted, stored at −80 °C, and did not undergo freeze-thaw cycles.

Blood was collected at inclusion (day 1), after three to five days, after seven to ten days, and subsequently at regular intervals for the duration of the hospital stay or until the patient’s death. A total of 128 plasma samples from 70 patients were included in this study. Of them, 103 samples were obtained from sequential sampling of 45 patients, while 25 samples were collected as inclusion-day samples from 25 patients who were either discharged from hospital or died within two days. The inclusion criteria are illustrated in Figure 1.

### 2.2. SARS-CoV-2 RNA Standard

SARS-CoV-2 virus, lineage B (European Virus Archive, ref. 008V-03893), was cultivated in Vero E6 cells (CRL-1586™, ATCC, Manassas, VA, USA), after which the supernatant was harvested at passage 4 through centrifugation at 300 g for 10 min at 4 °C and subsequently heat-inactivated for 30 min at 56 °C. SARS-CoV-2 RNA from the cell supernatant was titrated against the 20/146 WHO International Standard for SARS-CoV-2 RNA [52]. The supernatant was diluted 1:200 to match the concentration of the 20/146 standard (7.7 log10/mL) and further serially diluted 1:10 in five steps in UTM (Universal Transport Medium, Copan, Murrieta, CA, USA), aliquoted, and stored at −80 °C. The same standard curve dilutions were applied for both qRT-PCR and RT-ddPCR for quantification purposes. Limit of detection (LoD) determination was not performed.

### 2.3. Nucleic Acid Extraction

RNA was extracted from 200 µL of sample or standards using the QiaSymphony Virus/Pathogen kit (Qiagen, Hilden, Germany) using the cell-free protocol with bacteriophage MS2 RNA (Roche, Basel, Switzerland) added to each sample as an internal positive control. The elution volume was 60 µL. RNA eluates were stored at −80 °C for up to six months and subjected to no more than one freeze-thaw cycle prior to use.

### 2.4. Oligos

Oligos target the E-gene of SARS-CoV-2 [51] and the replicase gene of MS2 bacteriophage [53]. We utilized both targets for qRT-PCR detection but only the E-gene for RT-ddPCR detection. The fluorophores and quenchers for the E-gene oligos were consistent across both assays. Further specifications can be found in Table 1.

### 2.5. qRT-PCR Detection and Quantification

The qRT-PCR assays were performed immediately after extraction and before freezing and RT-ddPCR detection. All qRT-PCR assays were performed in triplicate, using 25 µL reaction volumes on an AriaDX PCR instrument (Agilent Technologies LDA, Penang, Malaysia). The reaction mixture comprised 400 nM E-gene primers, 200 nM E-gene probe, 300 nM MS2 primers, 100 nM MS2 probe, QuantiNova Pathogen Mastermix (Qiagen, Hilden, Germany), nuclease-free water, and 5 or 9 µL eluate. Thermal cycling conditions were set at 45 °C for 10 min for reverse transcription, followed by 95 °C for 5 min and then 40 cycles of 95 °C for 5 s and 60 °C for 30 s. All wells with a Cq (cycle quantification, equal to cycle threshold) value < 38 with sigmoid curves were considered positive.

Quantification was undertaken by developing a linear regression model using log10-transformed standard concentrations plotted against the corresponding Cq values in Python version 3.10.4 [54]. Subsequently, sample concentrations were calculated by interpolating the sample Cq values on the linear standard curve. The calculations underwent quality assurance both visually and by executing a subset in GraphPad version 10.2.0 (GraphPad Software, San Diego, CA, USA).

### 2.6. RT-ddPCR Detection

The RT-ddPCR assays were performed on the same eluates that were used for qRT-PCR detection, after being stored at −80 °C. All RT-ddPCRs were also performed in triplicate, using 22 µL reaction volumes, with 20 µL employed to generate droplets on a QX200 Auto Droplet Generator (BioRad, Hercules, CA, USA). The reaction included the Supermix, 20 U/µL, 15 nM DTT (1-step RT-ddPCR Advanced Kit for Probes, BioRad), 900 nM E-gene primers, 250 nM E-gene probe, nuclease-free water, and 5 or 9 µL eluate—the same as in qRT-PCR. Table 1 shows the primer and probe sequences used. Thermal cycling was carried out on ABI Veriti Thermal Cyclers (Applied Biosystems, Waltham, MA, USA) at 45 °C for 60 min for reverse transcription, followed by 95 °C for 10 min and then 40 cycles of 95 °C for 30 s and 60 °C for 60 s before enzyme deactivation at 98 °C for 10 min. The ramp rate was set to −2 °C/s. After amplification, droplets were analyzed using the QX200 Droplet Reader (BioRad). Wells with fewer than 10,000 droplets were excluded from further calculations, and the cut-off was set manually.

### 2.7. Verification of Linearity

The same standards were examined with both methods, and standard curves were generated using GraphPad version 10.2.0.

### 2.8. Statistical Analysis

We used McNemar’s test to preliminarily assess whether the two methods were equally effective, with the following null hypothesis: The number of samples negative on ddPCR and positive on qPCR is the same as the number of samples positive on ddPCR and negative on qPCR (Table 3). Linear regressions were conducted to determine whether the relationship between input copies/mL and Cq values or observed copies/mL was linear. A paired t-test was employed to examine if there was a difference in viral quantities for the qRT-PCR-positive/RT-ddPCR-positive samples, and an unpaired t-test was employed to evaluate any differences in viral quantities for the divergent results. All calculations were performed using GraphPad version 10.2.0, with two-sided *p* < 0.05 considered statistically significant.

For comparison of agreement and sensitivity, we analyzed the last qRT-PCR-positive and first qRT-PCR-negative sample from longitudinal samples using RT-ddPCR. All samples were quantified using both methods. After quantification via RT-ddPCR, the observed values (copies/mL) were used as standard values to quantify the plasma samples with qRT-PCR. This enabled a comparison of the registered viral quantities quantified using the two methods.

If qRT-PCR negative samples are positive using RT-ddPCR, this could suggest that RT-ddPCR is a more sensitive assay than qRT-PCR. Therefore, we constructed a 2 × 2 contingency table of the results to evaluate the relationship between the number of qRT-PCR-positive/RT-ddPCR-negative samples and qRT-PCR-negative/RT-ddPCR-positive samples.

To determine if RT-ddPCR registered higher viral quantities than qRT-PCR, which could indicate higher sensitivity, we compared the viral quantities of samples determined positive with both qRT-PCR and RT-ddPCR. We also compared the viral quantities of qRT-PCR-negative/RT-ddPCR-positive samples to those of qRT-PCR-positive/RT-ddPCR-negative samples. If RT-ddPCR is a more sensitive method than qRT-PCR, we would expect equal or lower viral quantities in qRT-PCR-negative/RT-ddPCR-positive samples compared to qRT-PCR-positive/RT-ddPCR-negative samples.

## 3. Results

In this study, we performed RT-ddPCR and qRT-PCR on 128 plasma samples from 70 hospitalized COVID-19 patients and evaluated the results.

First, we calculated the mean and standard deviations (SDs) for the observed copies/mL of the standard (Table 2) and conducted linear regressions on the standard curves for both qRT-PCR and RT-ddPCR. Both analytical methods demonstrated a linear relationship between expected virus copy number and the Cq value or the observed virus copy number (Figure 2) when performed on the same plasma tubes.

Subsequently, we summarized the discrepancies between the two methods (Table 3). A total of 89 out of 128 samples yielded concordant results regardless of the method used, while 39 out of 128 samples showed divergent results depending on the method. Seven more qRT-PCR negative samples were determined to be positive based on RT-ddPCR than vice versa, but this difference was not statistically significant (McNemar’s test; Χ^2^ = 0.9231, *p* = 0.3367).

**Table 3 viruses-17-00772-t003:** A 2 × 2 contingency table with sample results from qRT-PCR and RT-ddPCR.

	Positive RT-ddPCR	Negative RT-ddPCR	Total
Positive qRT-PCR	29	16	45
Negative qRT-PCR	23	60	83
Total	52	76	128

We then applied the observed copies/mL from RT-ddPCR (Table 2) as standard values for qRT-PCR quantification and quantified the virus RNA in plasma using qRT-PCR. The two methods registered significantly different viral quantities (log10 copies/mL) in the paired positive samples (*n* = 29, Table 3) (paired *t*-test, two-tailed, *p* = 0.0026), with RT-ddPCR often resulting in higher or similar viral quantities compared to qRT-PCR (Figure 3).

Finally, we compared the viral quantities of samples determined to be positive with qRT-PCR and negative with RT-ddPCR to the viral quantities of samples determined to be negative with qRT-PCR and positive with RT-ddPCR (Figure 3). The viral quantities were significantly different (unpaired *t*-test, two-tailed, *p* = 0.0075), but the viral quantities (median 652, IQR 489) in qRT-PCR-negative/RT-ddPCR-positive samples were *not* lower or equal to the viral quantities (median 329.9, IQR 361) in qRT-PCR-positive/RT-ddPCR-negative samples; rather, it was the opposite (Figure 4). 

## 4. Discussion

We analyzed 128 plasma samples from 70 inpatients diagnosed with COVID-19. Among them, 29 samples were determined to be PCR-positive using both methods, while 60 samples were PCR-negative across both methods. Sixteen samples were determined to be negative with RT-ddPCR but positive with qRT-PCR, whereas 23 samples were determined to be negative based on qRT-PCR but positive based on RT-ddPCR.

The viral quantities in this study were higher when quantified using RT-ddPCR than with qRT-PCR, which may suggest RT-ddPCR as a more sensitive method. Nevertheless, if this were indeed the case, one would anticipate the viral quantities in qRT-PCR-negative/RT-ddPCR-positive samples to be equal to or lower than the viral quantities in qRT-PCR-positive/RT-ddPCR-negative samples. In reality, we observed the opposite: the viral quantities in qRT-PCR-positive/RT-ddPCR-negative samples were equal to or higher than the viral quantities in qRT-PCR-negative/RT-ddPCR-positive samples. Hence, we cannot assert that RT-ddPCR is a more sensitive method than qRT-PCR. However, we can conclude that the methods provide comparable results.

We used the same sample eluates for assessing both methods to minimize bias from differences in the extraction process. The eluate for the qRT-PCR was used immediately after extraction but was frozen, carefully thawed on ice, and kept on ice before performing RT-ddPCR. Some degradation of the eluate before conducting the RT-ddPCR might explain why 16 samples were determined to be negative based on RT-ddPCR but positive based on qRT-PCR. However, as all eluates were treated similarly and the 16 samples came from different eluate batches and were run on different RT-ddPCR plates, we believe that the likelihood of degradation is low.

As all eluates were analyzed with qRT-PCR before being analyzed with RT-ddPCR, we obtained the necessary information on extraction efficiency and potential inhibitors from the MS2 qRT-PCR results and deemed a rerun of this by RT-ddPCR unnecessary. Consequently, MS2 oligos were not added to the RT-ddPCR reaction mixture, and this decision should not affect the RT-ddPCR results for E-gene detection.

We chose to follow procedures relevant to a clinical diagnostic routine laboratory. This involved using RNA as the input template rather than cDNA, which would require additional work and increase the cost and time of the analysis, as the mastermixes already contain RT. As a result, a bias between the two methods may have been introduced due to differing RT-steps (10 min for qRT-PCR versus 60 min for RT-ddPCR) and transcription efficiency [45]. The much shorter RT step in qRT-PCR could lead to less transcription of viral RNA, resulting in a higher Cq value or a negative result. This may explain why higher viral quantities are observed in RT-ddPCR compared to qRT-PCR.

The choice of extraction method may influence the output [55], but as we used the same automated routine instrument, protocol, and reagent lot on all extractions, we believe our comparison of RT-ddPCR to qRT-PCR to not be affected by this.

Our results do not completely concur with some studies performed on nasopharyngeal swabs, which concluded that RT-ddPCR is a more sensitive method than qRT-PCR [42,43,46,48,49,56,57,58]. Other studies partially agree with ours, concluding that RT-ddPCR can be equally or more sensitive than qRT-PCR [39,45,59]. However, results from previous studies are not directly comparable, as they primarily utilized other oligosets than we did in the present study, which influences the assay’s sensitivity [45,50,60,61,62,63,64,65,66]. Some studies exhibit bias by exclusively examining equivocal or qRT-PCR-negative/RT-ddPCR-positive samples [46,48,49] or by employing a very low sample size [56]. Only one study compares qRT-PCR and RT-ddPCR on clinical samples (nasopharyngeal swabs) using E-gene oligonucleotides, albeit with pooled samples [42].

Although this study could not establish that RT-ddPCR is more sensitive than qRT-PCR for detecting SARS-CoV-2 E-gene RNA in plasma, RT-ddPCR has other advantages. As qRT-PCR quantification can vary significantly depending on instrumentation, reagents, and standard curves [67,68], RT-ddPCR, with its high accuracy and lack of requirement for a standard curve, improves comparability between laboratories [69,70]. However, digital PCR is not completely free from sources of error, e.g., a high risk of cross-contamination and the significant impact of pipetting inaccuracies on copy number measurements [71,72]. Other considerations include the higher cost, longer hands-on time, instruments, and greater technical skill required for RT-ddPCR compared to qRT-PCR.

## Figures and Tables

**Figure 1 viruses-17-00772-f001:**
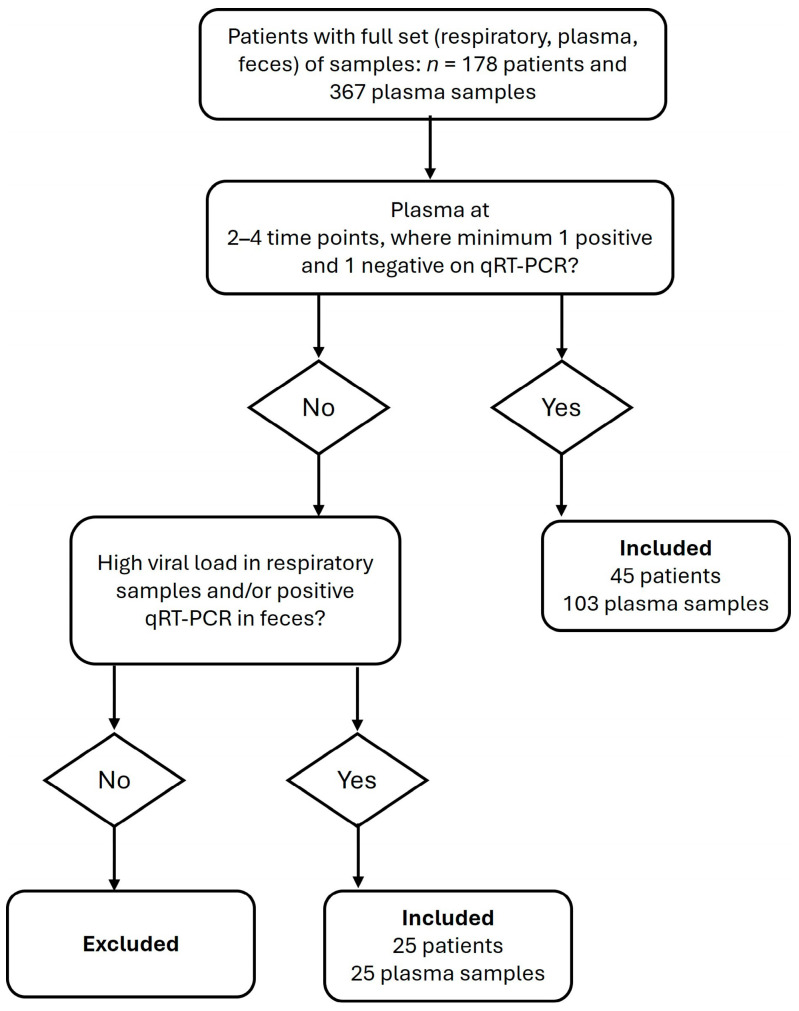
Flow chart from top to bottom, describing sample selection criteria.

**Figure 2 viruses-17-00772-f002:**
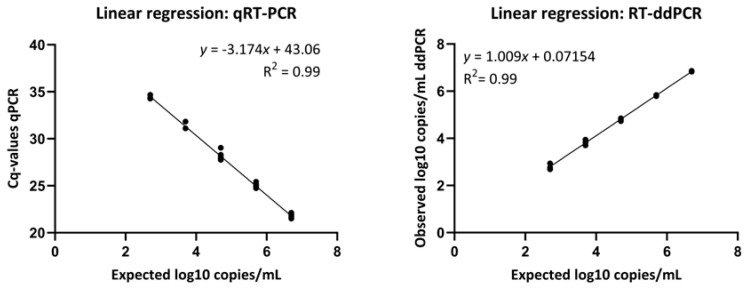
Standard curves for SARS-CoV-2 E-gene qRT-PCR and RT-ddPCR, generated through simple linear regression, where each dot represents one replicate. The standards are the same in both panels, as is their expected concentration in log10 copies/mL. The left panel displays how the standard’s expected concentration correlates with Cq values when analyzed using qRT-PCR. The right panel displays how the standard’s expected concentration correlates with observed concentration when analyzed with RT-ddPCR.

**Figure 3 viruses-17-00772-f003:**
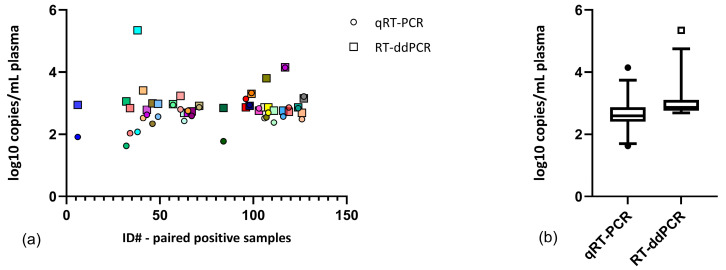
(**a**) XY scatterplot of all samples determined to be positive using both methods (*n* = 29), with the sample number on the x-axis and the concentration (log 10 copies/mL) on the y-axis. Each pair of results (RT-ddPCR result versus qRT-PCR result for one sample) is represented by the same color. (**b**) Box and whisker plot showing the mean and 5–95% percentile of positive results based on RT-ddPCR versus qRT-PCR.

**Figure 4 viruses-17-00772-f004:**
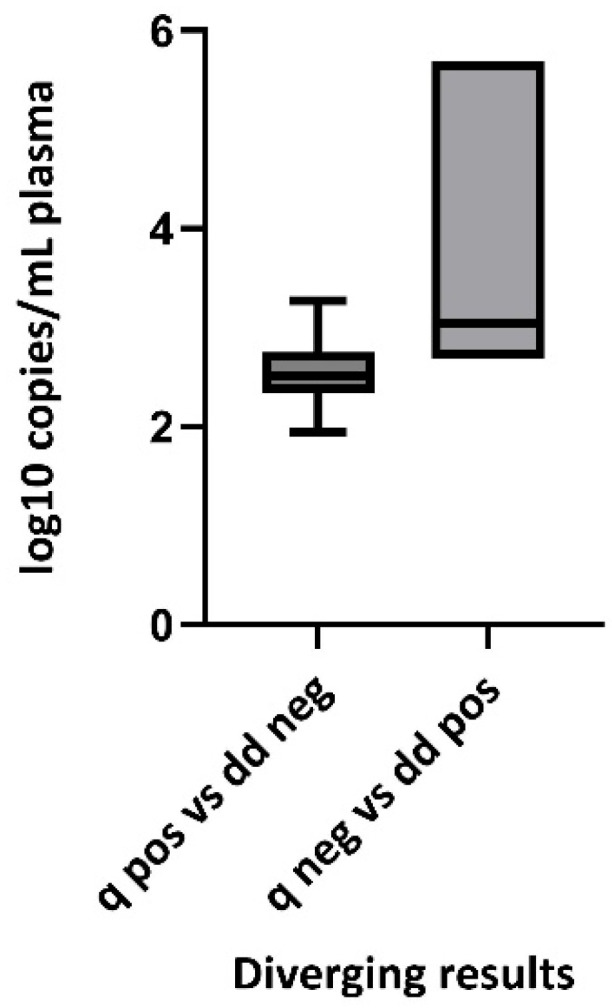
Box and whisker plot showing mean and 5–95% percentile of disconcordant samples; qRT-PCR-positive (q pos) versus RT-ddPCR-negative (dd neg) and qRT-PCR-negative (q neg) versus RT-ddPCR-positive (dd pos), n = 39.

**Table 1 viruses-17-00772-t001:** Oligosets used in qRT-PCR (reverse transcriptase real-time PCR) and ddPCR (reverse transcriptase digital droplet PCR).

Target	Primer/Probe	Sequence (5′-3′)	Coordinates *
E-gene (SARS-CoV-2)	Forward	ACAGGTACGTTAATAGTTAATAGCGT	26,269–26,294
	Reverse	ATATTGCAGCAGTACGCACACA	26,381–26,360
	Probe	FAM-ACACTAG/ZEN/CCATCCTTACTGCGCTTCG-IBFQ	26,332–26,357
Replicase gene (MS2)	Forward	TGCTCGCGGATACCCG	3169–3184
	Reverse	AACTTGCGTTCTCGAGCGAT	3229–3210
	Probe	HEX-ACCTCGGGTTTCCGTCTTGCTCGT-BBQ	3186–3209

* Coordinates according to SARS-CoV-2 (GenBank accession number MN908947.3) or bacteriophage MS2 (GenBank accession number V00642). FAM, 6-carboxyfluorescein; HEX, hexachloro-fluorescein; ZEN, internal ZEN quencher; IBFQ, Iowa Black Fluorescent Quencher; BBQ, BlackBerry Quencher. Primers were purchased from TibMolBio (Berlin, Germany), and TaqMan hydrolysis probes were purchased from Integrated DNA Technologies (Coralville, IA, USA).

**Table 2 viruses-17-00772-t002:** Comparison of expected copies/mL and observed copies/mL by RT-ddPCR and Cq value using qRT-PCR for the SARS-CoV-2 (severe acute respiratory syndrome coronavirus 2) WHO (World Health Organization) standard.

Dilution Factor	Expected * log10 copies/mL	Observed log10 copies/mL (Mean ± SD)	Cq Value (Mean ± SD)
10^−1^	6.7	6.85 ± 0.02	21.77 ± 0.20
10^−2^	5.7	5.82 ± 0.03	24.98 ± 0.17
10^−3^	4.7	4.79 ± 0.06	28.14 ± 0.30
10^−4^	3.7	3.82 ± 0.12	31.35 ± 0.42
10^−5^	2.7	2.8 ± 0.13	34.44 ± 0.22

* Expected copies/mL is calculated from the instructions for use for the First WHO International Standard for SARS-CoV-2 RNA, NIBSC code 20/146.

## Data Availability

The original contributions presented in this study are included in the article/Appendix A. Further inquiries can be directed to the corresponding authors.

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
