# Peer review of "Droplet Digital PCR or Real-Time PCR as a Method for Quantifying SARS-CoV-2 RNA in Plasma—Is There a Difference?"

_viruses, 2025, doi:10.3390/v17060772_

Round 1
Reviewer 1 Report
Comments and Suggestions for Authors
My comments are as follows:
- How the sample size was determined. Is it enough for the conclusion with statistical sounds?
- How the Limit of Detection (LoD) of each assay was determined? Just serail dilution of the template? It is recommended to apply a static method, for example, probit regression to acquire a more accurate value of LoD.
- As a common sense, the LoD of TaqMan real time PCR should be lower than 200 copies/ml, could the authors clarify the reason for their higher value of LoDs? And also, I found in Table 2, an extremely higher variation in the Observed copies/ml, comparing to the extremely lower variations in Ct-Value. Could I infer from these results that both the procedures of qPCR and RT-ddPCR in this work were not set to optimal conditions?
Author Response
Thank you for taking the time to review our manuscript; we greatly appreciate your effort. Here are our responses:
Comments 1: How the sample size was determined. Is it enough for the conclusion with statistical sounds?
Response 1: Thank you for your valuable comments. In addition to our experience in clinical validation, we planned this assay validation according to this publication: https://pubmed.ncbi.nlm.nih.gov/19415949/, which recommends at least 50 positive and 50 negative results for qualitative test performance according to CSLI guidelines. For quantitative tests, a minimum of 40 samples is often suggested. The requirements are met with RT-ddPCR as the reference method.
Comments 2: How the Limit of Detection (LoD) of each assay was determined? Just serail dilution of the template? It is recommended to apply a static method, for example, probit regression to acquire a more accurate value of LoD.
Response 2: We have not determined Limit of Detection. The only reason for analysing the standards is for quantification. Our main aim for this study was to evaluate qualitative test performance (virus RNA present/not present), not to establish the LoD. We added a clarification of this in line 64 and 93-94.
Comments 3: As a common sense, the LoD of TaqMan real time PCR should be lower than 200 copies/ml, could the authors clarify the reason for their higher value of LoDs? And also, I found in Table 2, an extremely higher variation in the Observed copies/ml, comparing to the extremely lower variations in Ct-Value. Could I infer from these results that both the procedures of qPCR and RT-ddPCR in this work were not set to optimal conditions?
Response 3: The LoD of the qRT-PCR is not presented but is for sure less than 200 copies/mL. The lowest standard analysed (2.8 log10 copies/mL) give Ct-values of about 34. The LoD is probably about 1:10 lower than this (ap 1,8 log10 copies/mL), based on experience. The higher variations in Observed copies/mL compared to Ct-value is because Ct-values are log10 transformed. We have log10 transformed the Observed copies/mL and their SD for an easier comparison and updated Table2. The procedures of qPCR and RT-ddPCR were tested and validated before the experiment and are set to optimal conditions.
Reviewer 2 Report
Comments and Suggestions for Authors
--Only 29 out of 128 samples gave consistent positive as confirmed by both methods. 39 out of 128 samples gave divergent results. This is concerning. Maybe a different extraction methods should have been tested.
--It is unclear why 10 min RT was used with regular RT-PCR while 60 min was used with RT-ddPCR. I am not sure if this caused some of the discrepancy.
--Two primer sets (E-gene and MS2) were used with regular PCR but only E-gene was used or reported for RT-ddPCR. It is uncleared if the use of MS2 would have given similar results between the two PCR approaches. Basically saying the inability to confirm RT-ddPCR is better is caused by the primers and probe selections.
--While long COVID was a mentioned as one of the reasons why testing RNA in plasma is needed, only up to 10 days of samples were collected. It is unclear if the method can track long COVID if the two methods are not concordant.
--It is unclear if the 128 samples (from 70 patients) showed any trend of viral RNA drop over time. Maybe the author can add more information?
--The conclusion that "Although this study could not establish that RT-ddPCR is more sensitive than qRT-PCR for detecting SARS-CoV-2 E-gene RNA in plasma." is correct (unfortunately). Given the extra labor and cost associated with digital RT-PCR, it is not a reliable method for quantify RNA.
Author Response
Thank you for taking the time to review our manuscript; we greatly appreciate your effort. Here are our responses:
Comments 1: Only 29 out of 128 samples gave consistent positive as confirmed by both methods. 39 out of 128 samples gave divergent results. This is concerning. Maybe a different extraction methods should have been tested.
Response 1: Thank you for your insightful suggestions. We agree that it would have been interesting to see if a different extraction method would give other results. However, our aim was not to compare different extraction methods but to compare the two methods as they would have been used in our daily diagnostic routine. The instrument used have been validated for clinical diagnostic use.
Comments 2: It is unclear why 10 min RT was used with regular RT-PCR while 60 min was used with RT-ddPCR. I am not sure if this caused some of the discrepancy.
Response 2: This is based on the manufacturers’ recommendations. The qRT-PCR method is validated for clinical diagnostic use with a 10-minute RT step. We discuss this further in the discussion (line 245-254).
Comments 3:Two primer sets (E-gene and MS2) were used with regular PCR but only E-gene was used or reported for RT-ddPCR. It is uncleared if the use of MS2 would have given similar results between the two PCR approaches. Basically saying the inability to confirm RT-ddPCR is better is caused by the primers and probe selections.
Response 3: MS2-oligos is mainly used as an extraction- and inhibition control, after adding the bacteriophage MS2-RNA to all samples before extraction. As we have used the same eluates for both qPCR and ddPCR, and also know that ddPCR is theoretically less susceptible to inhibitors, there is no need in repeating this PCR when running the ddPCR. The E-gene oligos are the same in both assays. It is highly unlikely that the inability to confirm RT-ddPCR is caused by the lack of MS2 oligos in the reaction mixture. We have added information about this under “Discussions”.
Comments 4:While long COVID was a mentioned as one of the reasons why testing RNA in plasma is needed, only up to 10 days of samples were collected. It is unclear if the method can track long COVID if the two methods are not concordant.
Response 4: Yes, we agree. Our observational study is however not about long COVID, but the methods that we compare our results with have been used in long COVID studies.
Comments 5: It is unclear if the 128 samples (from 70 patients) showed any trend of viral RNA drop over time. Maybe the author can add more information?
Response 5: Thank you for this comment. This is irrelevant for this method comparison, but we plan to publish data on this in another paper.
Comments 6: The conclusion that "Although this study could not establish that RT-ddPCR is more sensitive than qRT-PCR for detecting SARS-CoV-2 E-gene RNA in plasma." is correct (unfortunately). Given the extra labor and cost associated with digital RT-PCR, it is not a reliable method for quantify RNA.
Response 6: Thank you for finding our conclusion correct.
Reviewer 3 Report
Comments and Suggestions for Authors
The study compares two PCR-based nucleic acid detection approaches. Authors conducted a useful study and methods were compared using clinical samples. Revisions are needed before I recommend for publication.
- "A total of 128 plasma samples from 70 patients" are used in the study. Please clarify how many plasma samples per patient were included.
- Authors report discrepancies in both methods. It is not clear whether the samples positive by qPCR and negative by ddPCR were the same or different samples (for example, if sample X was qPCR positive, was the same sample ddPCR as well?)
- Authors stated that "We utilized both targets for qRT-PCR detection, but only the E-gene for RT-ddPCR detection." why both targets are not used for RT-ddPCR as well?
- Could it be useful to test the same samples by any other reliable method such as antibody tests for better conclusions?
Author Response
Thank you for taking the time to review our manuscript; we greatly appreciate your effort. Here are our responses:
Comments 1: "A total of 128 plasma samples from 70 patients" are used in the study. Please clarify how many plasma samples per patient were included.
Response 1: Thank you for these suggestions. We agree that this needs clarification. We have updated Figure 1 and line 78-80.
Comments 2: Authors report discrepancies in both methods. It is not clear whether the samples positive by qPCR and negative by ddPCR were the same or different samples (for example, if sample X was qPCR positive, was the same sample ddPCR as well?)
Response 2: These are the same sample and the same eluates. We have added an extra line describing this (line 129), in addition to the already existing information in the Discussions. We have also tried to visualize this in figure 3a, where each colour represents one eluate.
Comments 3: Authors stated that "We utilized both targets for qRT-PCR detection, but only the E-gene for RT-ddPCR detection." why both targets are not used for RT-ddPCR as well?
Response 3: We could have used both targets for both assays, but this would not have given us any more information. The reason why MS2 oligos are added to the qPCR assay is because MS2 bacteriophage RNA is added to each sample before extraction, as an extraction and inhibition control. As all eluates were analyzed by qPCR before ddPCR, we already had the information that a) extraction had worked and b) if the eluate contained inhibitors. We have added this information under “Discussions”.
Comments 4: Could it be useful to test the same samples by any other reliable method such as antibody tests for better conclusions?
Response 4: In our experience, PCR-based tests are more sensitive and with a lower variation than antibody-based tests. For better conclusions, we believe other oligosets, i.e ORF1ab, or more sensitive qPCR-methods, i.e nested PCR is of more use.